

# Spontaneous space closure after extraction of young first permanent molar. Retrospective cohort study

Heba Jafar Sabbagh[1], Ahmad Adnan Samara[2], Shoroog Hassan Agou[3], Jihan Turkistani[4], Manal Ibrahim Al Malik[5], Hadeel Abdulatief Alotaibi[6], Abrar Shimi Dhaifallah Alsolami[7] and Nada Othman Bamashmous[1]

[1] Pediatric Dentistry Department, Faculty of Dentistry, King Abdulaziz University, Jeddah, Saudi Arabia
[2] Pediatric Dentistry, Ministry of Health, Taif, Saudi Arabia
[3] Department of Orthodontics, Faculty of Dentistry, King Abdulaziz University, Jeddah, Saudi Arabia
[4] Department of Dental Services, King Abdulaziz Medical City, Ministry of National Guard Health Affairs, Jeddah, Saudi Arabia
[5] Dental Department, King Fahad Armed Forces Hospital, Jeddah, Saudi Arabia
[6] Department of Dental Services, East Nakb Primary Health Care Center, Ministry of Health, Taif, Saudi Arabia
[7] Saudi Board in Endodontics, Ministry of Health, Makkah, Saudi Arabia

Corresponding author
Nada Othman Bamashmous,
nobamashmous@kau.edu.sa

## ABSTRACT

**Background:** Early compromised first-permanent-molar (FPM) extraction can adversely affect occlusion. However, the current literature does not provide sufficient support for definitive early extraction outcome. This cohort study aimed to evaluate the pattern of post-FPM extraction spontaneous space closure and its pre-extraction indicators.

**Methods:** Patients of 7–13 years, with compromised FPM at three-tertiary-centers between 2015 and 2019 were retrospectively screened. Pre-extraction indicators were evaluated (extraction location, direction of the second permanent molar (SPM) long-axis, and SPM Demirjian development stage). Spontaneous space closure pattern was evaluated clinically and radiographically using the American Board of Orthodontics (ABO) grading system.

**Results:** In total, 112 FPMs (73-patients) were identified and five (4.5%) of the extraction sites showed favorable combined-ABO-scores. Distal direction of the SPM long-axis in the maxillary arch significantly decreased the relative risk (RR) of having a SPM rotation/alignment ABO score-1 ($P = 0.002$; RR = 0.64). For the mandibular arch, Demirjian stage D and E significantly decreased the RR of having ABO score-1 for spacing between the SPMs and second premolars ($P = 0.029$; RR:0.57, $P = 0.004$; RR:0.53, respectively) and vertical dimension ($P = 0.02$; RR = 0.53).

**Conclusions:** The direction of the SPM long-axis and its developmental stage are key indicators of the favorable outcome pattern of spontaneous space closure after FPM early extraction.

## INTRODUCTION

The first permanent molar (FPM) is the earliest permanent tooth to erupt in the oral cavity, making it particularly susceptible to caries (*Meer, 2012*). Moreover, recent studies have shown an increase in the prevalence of molar-incisor hypomineralization (MIH), which can facilitate bacterial penetration through exposed dentin, potentially leading to pulpal involvement and an elevated risk of FPM loss (*Bukhari et al., 2023*; *Reissenberger et al., 2022*).

The management of compromised FPM ranges from root canal therapy (*Dhafar et al., 2022*; *Marending, Attin & Zehnder, 2016*) to extraction (*Albadri et al., 2007*). Although root canal therapy in children is considered more conservative than extraction and was linked to good oral health related quality of life (*Bamashmous et al., 2024*), in many situations it requires adjunctive treatment that could be difficult to perform in younger people, such as crown lengthening or the application of conventional crowns (*Alkhalaf et al., 2020*). In addition, complete root canal therapy is challenging to conduct in cases of uncooperative patients or those who cannot withstand long-duration treatments (*American Academy of Pediatric Dentistry, 2022*; *Dhafar et al., 2022*; *Sivakumar & Gurunathan, 2019*). Moreover, some young and compromised FPMs have a poor prognosis, affecting the decision of clinicians regarding extraction (*Dhafar et al., 2022*). Some orthodontic studies have recommended that extracting the compromised FPM instead of sound premolars may be more beneficial (*Ong & Bleakley, 2010*).

Loss of permanent molars affects many aspects of individual's occlusion and oral health (*Saber et al., 2018*). It leads to malocclusion which is a well-known problem that presents a health issue, affecting the quality of life of young patients (*Dimberg, 2015*). Nevertheless, studies have suggested that the early extraction of compromised FPMs could reduce the drawbacks and problems in occlusion; however, this remains under consideration, as many variables can come into play (*Saber et al., 2018*). Moreover, FPM extraction timing and pattern must be patient-specific to achieve the best results (*Ong & Bleakley, 2010*). Treatment modality, current dentition, dentition developmental stage, presence of malocclusion, overall prognosis (*Cobourne, Williams & Harrison, 2014*), and child age could influence the FPM treatment decision. A systematic review of 11 studies, encompassing all languages and data up to 2017, concluded that the current literature lacks sufficient support for definitive outcomes due to various limitations in study design and methodology (*Saber et al., 2018*). Previous studies have not comprehensively evaluated key aspects related to spontaneous space closure after first permanent molar (FPM) extraction, such as the buccolingual inclination, rotation, root angulation of second permanent molars (SPM), and optimal spontaneous space closure. Methods and parameters reported in earlier research include using cephalometric radiographs to assess the angulation of the SPM longitudinal axis (*Telli & Aytan, 1989*), panoramic or bitewing radiographs to measure the distance between posterior teeth (*Jälevik & Möller, 2007*; *Telli & Aytan, 1989*), and clinical assessments using a perio-probe (*Teo, Ashley & Derrick, 2016*). The complexity of a comprehensive evaluation and the need for long-term follow-up have further hindered past assessments (*Saber et al., 2018*). For instance, a recent study was unable to address

these issues due to the absence of a definitive measurement tool for thoroughly evaluating space closure patterns (*Brusevold et al., 2022*).

Therefore, the aim of this study was to assess the pattern of spontaneous space closure following FPM extraction, focusing on the SPM and premolars, utilizing the American Board of Orthodontics (ABO) cast-radiograph model grading system as a guide. Additionally, the study explores pre-extraction factors and their correlation with the pattern of spontaneous space closure after FPM extraction.

## MATERIALS AND METHODS

### Study design

This retrospective cohort study adhered to the Strengthening of the Reporting of Observational Studies in Epidemiology (STROBE) guidelines. It was conducted at three referral centers (King Abdulaziz University Dental Hospital (UDH), King Fahad Armed Forces Hospital (KFAFH), and King Abdulaziz Medical Center (KAMC) in Jeddah City, Saudi Arabia; these hospitals have a diverse and heterogeneous patient population. UDH is an educational hospital that treats a widely heterogeneous population from different socioeconomic groups. KFAFH and KAMC treat military and national guard personnel and their families. Ethical approval was obtained from the Research Ethics Committee of King Abdulaziz University Faculty of Dentistry (#005-01-19), Research Ethics Committee of King Fahad Armed Forces Hospital (#REC 277), and the Institutional Review Board (IRB) of Ministry of National Guard at King Abdullah International Medical Research Center (#H-01-R-005). The inclusion criteria for selecting patients were:

- Normal healthy individuals
- Underwent FPM extraction at the age of 7–13 years,
- The presence of panoramic radiological images at the time of the extraction
- Patient with either completely unerupted SPM at the time of extraction or fully erupted SPM in full-occlusion at the time of examination (stage-4 classification according to *Carvalho, Ekstrand & Thylstrup (1989)*).

The exclusion criteria were as follows:

- Having undergone extraction of a tooth other than the FPM,
- Unavailable radiographs at the time of FPM extraction,
- Erupted or partially erupted SPM at the time of FPM extraction,
- Not fully erupted SPM at assessment time according to *Carvalho, Ekstrand & Thylstrup, (1989)* (stage 1: partially erupted occlusal surface; or stage 2 and 3: fully erupted occlusal surface with the crown exposed but not in-full occlusion) (*Alves et al., 2014*; *Carvalho, Ekstrand & Thylstrup, 1989*),
- Having undergone orthodontic treatment.

The sample size calculation was conducted using G*Power 3.1.9.7, employing linear multiple regression (Fixed model) with an effect size of 0.15, α error probability of 0.05,
sample power of 0.80, and two predictors. This resulted in an estimated sample size of 68 patients. Additionally, to account for the design effect (clustering), further adjustment was made using an online application named Sample Size Calculator-UK Samples (*UK Samples, 2024*, accessed 10 June 2023) with a confidence level of 95%, required precision of five, intracluster correlation of 0.3 (*Masood, Masood & Newton, 2015*), and estimated proportion of 4.67. This yielded a design effect of 1.60 and a sample size estimate of 68 patients and 109 teeth.

## Data collection tool

The data collection sheet was developed following a literature review (*Saber et al., 2018*).

It was divided into two sections. "Introduction" covered general patient and tooth information, including patient demographics (age at extraction, sex, nationality, and medical history) and details about the extracted tooth (side of extraction, date of extraction, and reason for extraction). "Materials and Methods" addressed clinical and radiographic parameters, both pre-extraction and post-extraction.

Pre-extraction parameters (at the time of extraction) included:

- The direction of the long axis of the SPM, in relation to the occlusal plane recorded according to *Langer et al. (2023)*. The long axis of the SPM was either mesial, perpendicular, or distally directed relative to the occlusal plane. Figure 1 illustrates this parameter.
- The stage of SPM development according to the Demirjian stage of development, which includes Stage D (completed crown formation), Stage E (early root formation and bifurcation), Stage F (root developed equal to or greater than the size of the crown), and Stage G (root developed but apical ends still not closed) (Fig. 2).

Post-extraction parameters (at assessment time). These parameters were evaluated after the SPM had fully erupted and were considered follow-up parameters. They included:

- Radiographic and cast evaluations. These were performed according to the American Board of Orthodontics (ABO) cast-radiograph model grading system (*American Board of Orthodontics, 2012*). The ABO grading system is a reliable and validated scale that assesses the quality of post-orthodontic treatment. In this study, we analyzed the dental cast using five of the ABO parameters (four clinical and one radiographic): (1) rotation/alignment: alignment and rotation of the SPMs and second premolars; (2) spacing: interproximal contact between the SPM and second premolar, and interproximal contact between premolars; (3) vertical dimension: the distance between the marginal ridge of the SPM and second premolar, to assess proper vertical positioning of the posterior teeth; (4) inclination: buccolingual inclination of the SPM. (5); and root angulation: the root angulation of the SPM with the second premolar on panoramic radiograph, which evaluates how perfectly the roots of the teeth are positioned relative to each other. Figure 3 illustrates the five ABO parameters. The root angulation and inclination parameters were used to assess proper occlusion in maximum intercuspation of the

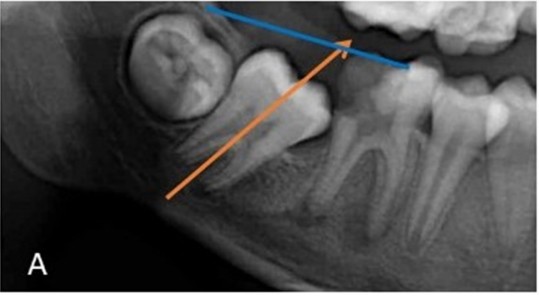 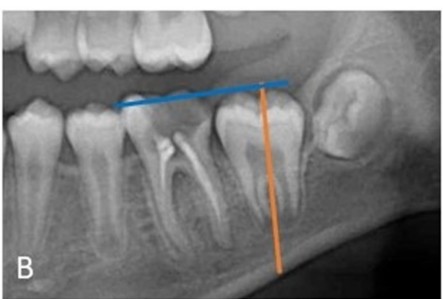 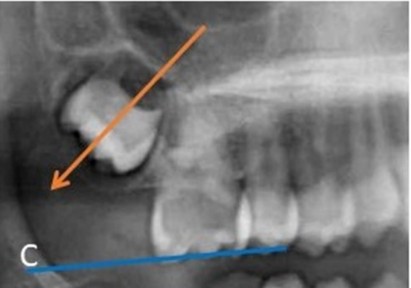

**Figure 1** (A) Long axis of the SPM is directed mesially to the occlusal plane; (B) long axis of the SPM is perpendicular to the occlusal plane; and (C) long axis of the SPM is directed distally to the occlusal plane.

SPM. The remaining five criteria were not assessed as they were not applicable in this context.

The rotation/alignment, interproximal spacing, and vertical dimension were evaluated using the ABO Measuring Gauge: "score 0" for values <0.5 mm and "score 1" for values ≥0.5 mm. Buccolingual inclination was scored as "score 0" for values <1 mm and "score 1" for values ≥1 mm. In assessing root angulation, if the roots of the maxillary and mandibular SPMs were parallel to the second premolar and oriented perpendicular to the occlusal plane, a score 0 was assigned. However, if the root was angled to the mesial or distal plane, a score 1 was assigned. The ABO grading system further groups score 1 to two groups: 0.5–1 and >1, which was not requisite in this study. A score 0 in rotation/ alignment, spacing, vertical dimension, inclination and root angulation indicates a favorable spontaneous space closure between the SPM and premolars.

## Method for data collection

The Information Technology Department was contacted to filter the electronic file system in the three centers to include pediatric patients who visited the centers between September 1, 2015, and August 30, 2019. The initial date (2015) was chosen because the information in the electronic filing system was available starting from that date. Data were screened, and cases of FPM extraction were identified according to the inclusion criteria. Among these, 73 patients who met the inclusion criteria were scheduled for appointments at pediatric dental clinics. The time elapsed between extraction and follow-up was recorded. Parents were asked to sign consent forms explaining the research objectives, methodology, and confidence. They were informed that the recruited information from files, patients, clinical examination, radiographs and casts would be used for research purposes and will be published. Additionally, any radiographic images used in the publication would not contain identifying patient information.

From the patient's file, a panoramic radiograph taken at the time of extraction was reviewed for pre-extraction parameters. Then, and a new panoramic radiograph was obtained at the assessment time. The radiographs were obtained according to the follow-up protocol of the American Academy of Pediatric Dentistry guideline, which allow for panoramic radiograph when required according to the clinical judgment for

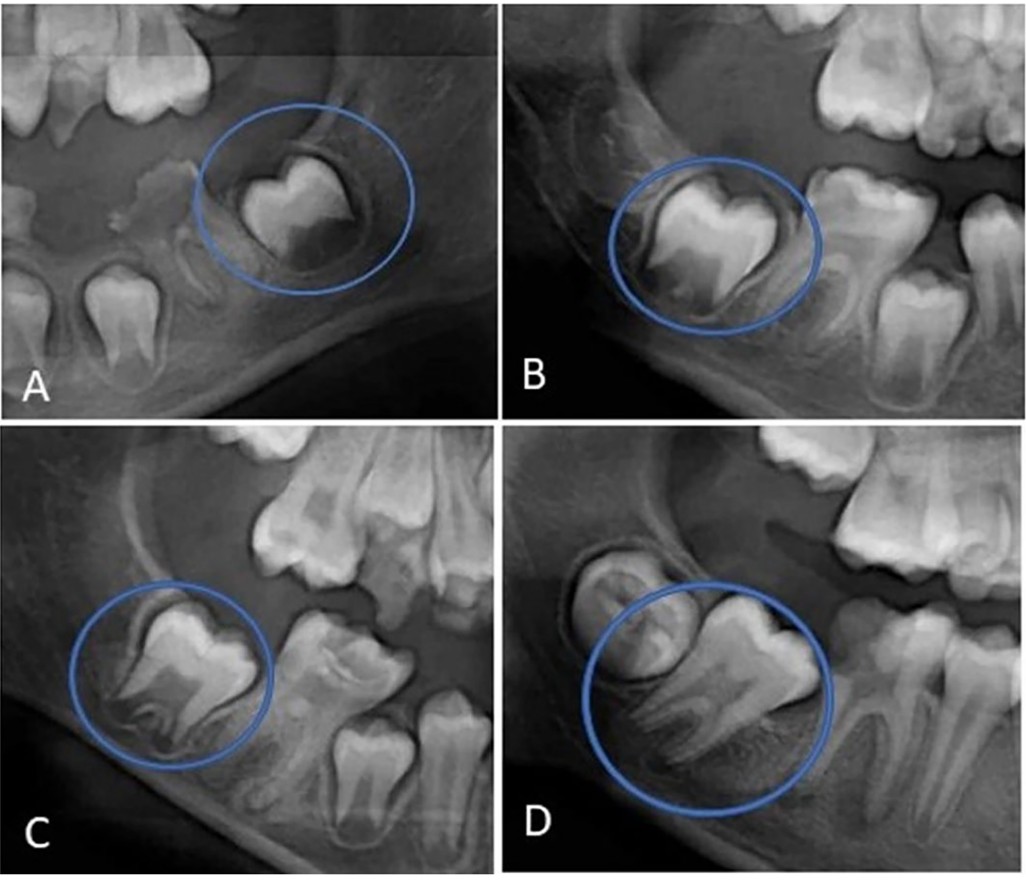

**Figure 2** (A) Demirjian stage D (completion of crown formation), (B) Demirjian stage E (early root formation and bifurcation), (C) Demirjian stage F (root developed equal or greater than the size of the crown), and (D) Demirjian stage G (root developed but apical ends).

monitoring dental development and relationships (*American Academy of Pediatric Dentistry, 2023*). In addition, an alginate impression was taken for the patient at the assessment time and poured onto the orthodontic casts for post-extraction parameters.

## Ascertainment and reliability

The data were collected by one main examiner (AS) and two data collectors (HA, AD) responsible for calling the patients and collecting data from the files. To improve reliability, only the main examiner was responsible for examining the radiographs and casts; however, to ascertain the measurements, the main examiner initially conducted case-radiographic evaluations for ten cases with the help of an orthodontic expert (SA). After they achieved a 100% agreement with the expert, intra-examiner reliability test was conducted for the main examiner (AS) within 2-month intervals for another ten cases (the kappa coefficient was 0.93). The main examiner was blinded from the child's sociodemographic data and pre-extraction parameters when measuring post-extraction parameters.

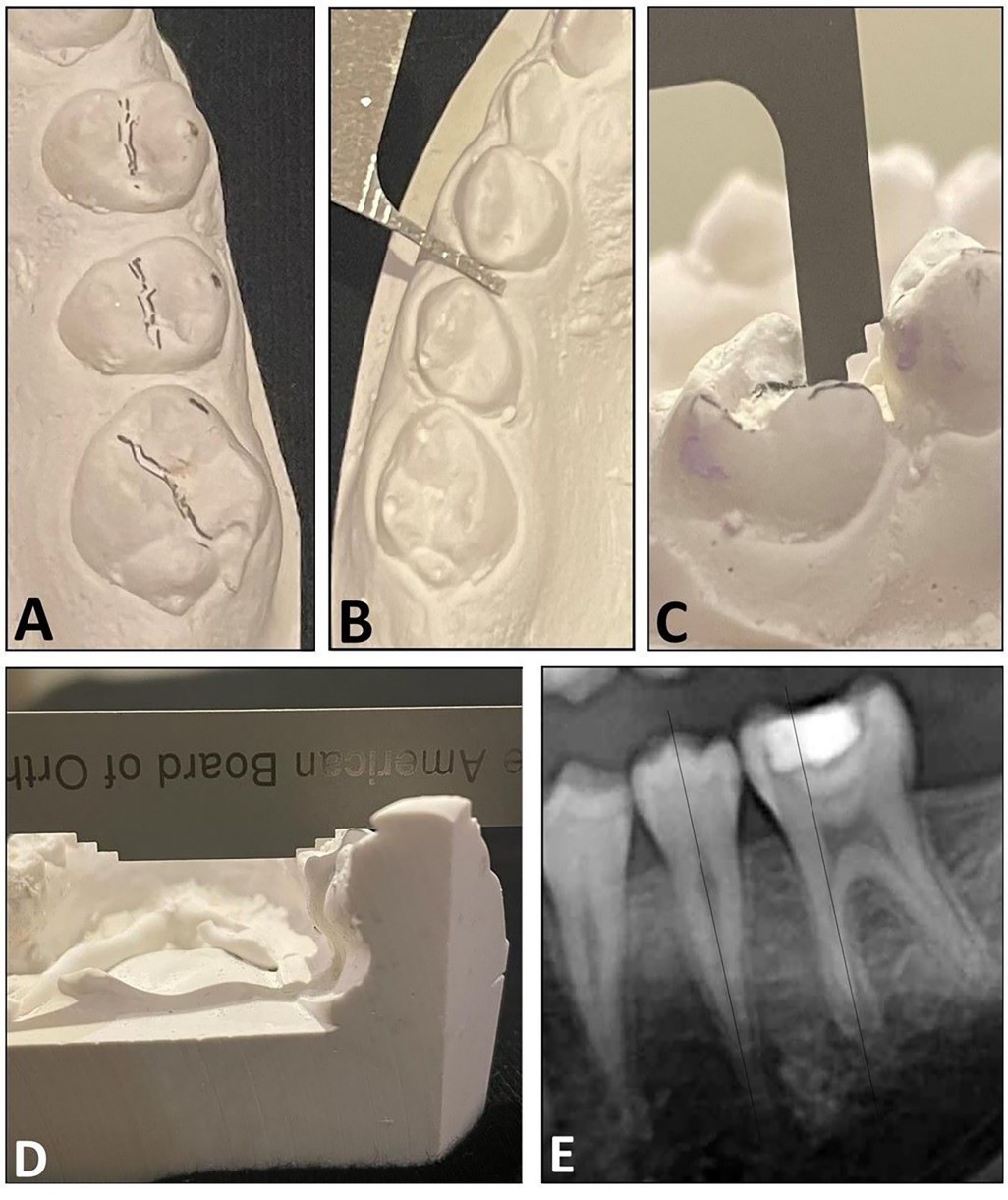

**Figure 3 The ABO cast-radiograph model grading system used to assess the pattern of alignment after early extraction of the first permanent molar.** (A) Rotation/alignment, (B) spacing, (C) vertical dimension, (D) inclination (buccolingual); and (E) root angulation.

## Statistical analysis

All data analyses were performed using IBM SPSS Statistics for Windows, version 23 (IBM Corp., Armonk, NY, USA). Descriptive statistics for continuous and categorical data were estimated as means, standard deviations, percentages, and frequencies. The chi-square test was used to compare the percentages/frequencies between groups. Relative risk (RR) and 95% confidence intervals (CI) were calculated to evaluate the probability of having non-favorable ABO grading score (score 1).

Multilevel linear regression-analysis with one-level structure comprising tooth number was conducted to assess the association between the mean combined ABO-cast measurements of FPM space closure pattern (dependent factor) and sex, arch of extracted FPM, Demirjian stage of SPM, and the direction of the long axis of SPM at pre-extraction time (independent factors). All ABO-cast parameters were combined and the scores ranged from 0 to 6; where 0 is the favorable and six is the least- favorable outcome. Statistical significance was set at 0.05.

## RESULTS

After screening all patients (aged 7–13 years) who underwent extraction of their FPM during the 4-year study period, a total of 192 patients were identified: 49/192 (25.5%) were from UDH, 95/192 (49.5%) were from KFAFH, and 48/192 (25%) were from KAMC. Of these, 73 patients who met the inclusion criteria and agreed to participate were included in the study, with 24/73 (32.9%) from UDH, 35/73 (47.9.3%) from KFAFH, and 14/73 (19.2%) from KAMC. The distribution of samples across centers was based on their total number of performed FPM extractions. In addition, 61.6% (45/73) of the recruited sample were females, which followed the female distribution of the total identified cases (122/192 (63.5%)); see Table 1.

The mean age of patients at the time of FPM extraction was 9.52 ± 1.5 years, ranging from 7 to 13 years. At the assessment time, the mean age was 14.20 ± 1.6 years. The time elapsed between extraction and assessment time ranged from 2 to 6 years, with a mean duration of 4.26 ± 0.9 years. Within the included cases ($n = 73$), the total number of FPM extractions was 112; of which 43/112 (38.4%) were right lower FPM (see Table 1).

Of the extracted FPMs ($n = 112$), 33/112 (29.5%) were maxillary and 79/112 (70.5%) were mandibular. At pre-extraction time, 33/112 (29.5%) of the adjacent SPMs were at Demirjian stage D and 54 (48.2%) at stage E. The direction of the long axis of the SPMs in the maxillary arch were either distal (28/112 (84.8%)) or perpendicular (5/112 (15.2%)) with none displaying a mesial direction. On the other hand, the direction of the long axis of the SPMs in the mandibular arch were distal, perpendicular, or mesial, in 1/112 (1.3%), 7/112 (8.9%), and 71/112 (89.8%) extractions, respectively (Table S1).

After complete eruption of the SPM at assessment time, ABO score 0 for spacing was observed in 66/112 cases (58.9%) between SPM and second premolar, and in 83/112 cases (74.1%) between premolars (see Table 2). However, there were five extraction sites that showed favorable ABO scores (score 0) for all clinical-radiographic parameters combined, accounting for 4.5% of the total sample. Among these cases, four were maxillary, four were at Demirjian stage D, and four had their SPMs long axis directed distally at pre-extraction time. The only mandibular extraction site that had favorable ABO score for all parameters was that which had the SPMs long axis directed perpendicular at the pre-extraction time.

ABO scores 0 for rotation/alignment of the SPM ($P = 0.009$), spacing between the SPM and second premolar, buccolingual inclination of the SPM and root angulation of the SPM ($P < 0.001$) were significantly more in the maxillary compared to the mandibular extraction sites. However, there were no statistically significant differences between the maxillary and the mandibular extraction sites in terms of the ABO scores of the vertical

**Table 1 Distribution of participants and extracted FPM among three medical centres according to sex and location (total number of participants = 73; number of extracted FPM = 112).**

| Variable | | Healthcare centre | | | Total |
|---|---|---|---|---|---|
| | | UDH n (%) | KFAFH n (%) | KAMC n (%) | n (%) |
| **Participants** | | | | | |
| Sex | Males | 8 (33.3) | 15 (42.9) | 5 (35.7) | 28 (38.4) |
| | Females | 16 (66.6) | 20 (57.1) | 9 (64.3) | 45 (61.6) |
| **Extracted FPM** | | | | | |
| No. of extracted FPM | 16 | 5 (17.2) | 7 (33.3) | 7 (11.3) | 19 (17.0) |
| | 26 | 4 (13.8) | 5 (23.8) | 5 (8.1) | 14 (12.5) |
| | 36 | 10 (34.5) | 3 (14.3) | 23 (37.1) | 36 (32.1) |
| | 46 | 10 (34.5) | 6 (28.6) | 27 (43.5) | 43 (38.4) |
| Extracted FPM location | Maxillary | 9 (31.0) | 12 (57.1) | 12 (19.4) | 33 (29.5) |
| | Mandibular | 20 (69.0) | 9 (42.9) | 50 (80.6) | 79 (70.5) |
| Total # teeth | | 29 (100) | 21 (100) | 62 (100) | 112 (100) |

Note:

FPM, First Permanent Molar; 16, upper right first permanent molar; 26, upper left first permanent molar; 36, lower left first permanent molar; 46, lower right first permanent molar; UDH, King Abdulaziz University Dental Hospital; KFAFH, King Fahad Armed Forces Hospital; KAMC; King Abdulaziz Medical Center.

dimension, the premolar's rotation/alignment, and spacing scores ($P$ = 0.513, 0.688, and 0.465, respectively). See Table 2.

In maxillary FPMs extraction, distal SPM long axis significantly decreased the risk of having an SPM rotation/alignment ABO score 1 ($P$ = 0.002; RR = 0.64). In addition, Demirjian stage D and E, and distal SPM long axis decreased the RR of having spacing and buccolingual inclination ABO score 1 of the SPM compared to Demirjian stage F and G and perpendicular SPM long axis, respectively. However, this association was not statistically significant ($P$ > 0.05).

Nevertheless, extracting the FPM in the mandibular arch during early Demirjian stage (stage D & E) significantly decreased the RR of having ABO score 1 for spacing between the SPMs and second premolars ($P$ = 0.029, RR = 0.57 for Demirjian stage D; and $P$ = 0.004, RR = 0.53 for Demirjian stage E) and vertical dimension ($P$ = 0.02; RR = 0.53 for Demirjian stage E). Additionally, although not statistically significant, having a perpendicular SPM long axis at pre-extraction time decreased the RR of having ABO score 1 for spacing ($P$ = 0.1, RR:0.22) and vertical dimension ($P$ = 0.60; RR: 0.52). On the other hand, the RR of ABO score 1 for the space between the premolars increased with Demirjian stage D and distal SPM long axis at pre-extraction time; however, these differences were only significant for the direction of SPM long axis ($P$ < 0.001, RR = 3.74). See Table 3.

In addition, multilevel linear regression analysis was conducted to evaluate the relationship between sex, pre-extraction Demirjian Stage of SPM and the direction of the long axis of SPM before FPM extraction (independent factors), and combined ABO-cast measurements of FPM space closure (dependent factor) in the maxilla, mandible and both

**Table 2 ABO scores for extraction sites after complete eruption of the second permanent molars distributed according to the arch of extraction.**

| ABO-scores | | | Maxillary (%) | Mandibular (%) | Total (%) | P value[c] |
|---|---|---|---|---|---|---|
| I. Rotation/alignment | SPM[$] | Score-0 | 10 (30.3) | 21 (26.6) | 50 (44.6) | 0.009* |
| | | Score-1 | 23 (69.7) | 58 (73.4) | 62 (55.4) | |
| | Second premolar[$] | Score-0 | 21 (63.6) | 29 (36.7) | 50 (44.6) | 0.688 |
| | | Score-1 | 12 (36.4) | 50 (63.3) | 81 (55.4) | |
| II. Spacing | Bet. SPM and second premolar[$] | Score-0 | 29 (87.9) | 37 (46.8) | 66 (58.9) | <0.001* |
| | | Score-1 | 4 (12.1) | 42 (53.2) | 46 (41.1) | |
| | Bet. premolars[$] | Score-0 | 26 (78.8) | 57 (72.2) | 83 (74.1) | 0.465 |
| | | Score-1 | 7 (21.2) | 22 (27.8) | 29 (25.9) | |
| III. Vertical dimension | Marginal ridge bet. second premolar and SPM[$] | Score-0 | 21 (63.6) | 45 (57.0) | 66 (58.9) | 0.513 |
| | | Score-1 | 12 (36.4) | 34 (43.0) | 46 (41.1) | |
| IV. Inclination | Bucco-lingual inclination of SPM[#] | Score-0 | 22 (66.7) | 3 (3.8) | 25 (22.3) | <0.001* |
| | | Score-1 | 11 (33.3) | 76 (96.2) | 87 (77.7) | |
| V. Root angulation | Of SPM[##] | Parallel | 18 (54.5) | 3 (3.8) | 21 (18.7) | <0.001* |
| | | Not parallel | 15 (45.5) | 76 (96.2) | 91 (81.3) | |

Notes:
SPM, Second Permanent Molar; ABO, American Board of Orthodontics; Bet, between.
[$] Score 0 for parameters <0.5 mm and 1 for parameters ≥0.5 mm.
[#] Buccolingual inclination of SPM: Score parameter: 0 for parameters <1 mm and 1 for parameters ≥1 mm.
[##] Radiographic scoring for root angulation: parallel roots and not parallel for nonparallel roots, and for roots contacting adjacent roots.
[c] Chi square test.
* P value significant at 0.05.

arches. In the maxillary arch, mesial SPM long axis was not observed. Consequently, we included distal and perpendicular SPM long axes in the analysis of the maxillary arch. In addition, age (a confounder variable) was excluded from the regression analysis because of the significant collinearity that existed between age and Demirjian stage.

There was no statistically significant association between any of the independent variables and the combined ABO-cast measurements in the maxilla and mandible arches separately. However, the direction of the long axis of SPM showed a statistically significant relationship with the combined ABO-cast measurements when both arches were considered together ($P = 0.025$ for perpendicular angulation and $P < 0.001$ for distal angulation of SPM compared to mesial angulation of SPM). Nevertheless, distal direction of the long axis of SPM showed a decreased tendency of having a favorable outcome of combined ABO-cast measurements compared to the perpendicular angulation of the SPM in the maxillary arch ($P = 0.48$). Additionally, in the mandibular arch, perpendicular angulation of SPM showed a decreased tendency of having a favorable outcome of combined ABO-cast measurements compared to mesial angulation, ($P = 0.08$) and distal compared to mesial ($P = 0.22$). However, these relationships were not statistically significant. See Table 4.

## DISCUSSION

The treatment of compromised FPMs in children often has an uncertain prognosis. While extraction is frequently proposed (*Baakdah et al., 2022*), there is considerable controversy

**Table 3** Association between an ABO score of '1' and pre-extraction parameters expressed as relative risk (RR) distributed according to the arch of extraction.

| ABO-scores | | ABO score | | | | | |
|---|---|---|---|---|---|---|---|
| | | Maxillary | | | Mandibular | | |
| | | Score-0 n (%) | Score-1[R] n (%) | P value[c], RR (95% CI) | Score-0 n (%) | Score-1[R] n (%) | P value, RR (95% CI) |
| **I. Rotation/Alignment** | | | | | | | |
| **SPM [$]** | | | | | | | |
| Demirjian stage | Stage D | 5 (50) | 7 (30.4) | 0.94, 0.97 [0.41–2.3] | 4 (19) | 17 (29.3) | 0.42, 1.16 [0.81–1.65] |
| | Stage E | 3 (30) | 13 (56.5) | 0.83, 1.35 [0.64–2.88] | 11 (52.4) | 27 (46.6) | 0.93, 1.02 [0.71–1.44] |
| | Stage F/G | 2 (20) | 3 (13) | | 6 (28.6) | 14 (24.1) | |
| Direction of the SPM long axis | Distal | 10 (100) | 18 (78.3) | 0.002*, 0.64 [0.49–0.85] | 1 (4.8) | 0 | 0.37, 0.34 [0.03–3.72] |
| | Perpendicular | 0 | 5 (21.7) | | 2 (9.5) | 5 (8.6) | 0.86, 0.96 [0.59–1.56] |
| | Mesial | 0 | 0 | | 18 (85.7) | 53 (91.4) | |
| Total | | 10 (100) | 23 (100) | | 21(100) | 58 (100) | |
| **Second premolar** | | | | | | | |
| Demirjian stage | Stage D | 8 (38.1) | 4 (33.3) | 0.52, 1.67 [0.24–11.45] | 10 (34.5) | 12 (23.5) | 0.97, 0.99 [0.57–1.72] |
| | Stage E | 9 (42.9) | 7 (58.3) | 0.40, 2.19 [0.35–13.76] | 10 (34.5) | 28 (54.9) | 0.19, 1.34 [0.86–2.08] |
| | Stage F/G | 4 (19) | 1 (8.3) | | 9 (31.0) | 11 (21.6) | |
| Direction of the SPM long axis | Distal | 18 (85.7) | 10 (83.3) | 0.69, 1.43 [0.24–8.38] | 1 (3.4) | 0 | 0.45, 0.4 [0.03–4.39] |
| | Perpendicular | 3 (14.3) | 2 (16.7) | | 2 (6.9) | 6 (11.8) | 0.45, 1.18 [0.76–1.83] |
| | Mesial | 0 | 0 | | 26 (89.7) | 45 (88.2) | |
| Total | | 21 (100) | 12 (100) | | 29 (100) | 51 (100) | |
| **II. Spacing** | | | | | | | |
| **Between SPM and second premolar** | | | | | | | |
| Demirjian stage | Stage D | 11 (37.9) | 1 (25.0) | 0.67, 0.41 [0.03–5.43] | 11 (39.7) | 10 (23.8) | 0.04*, 0.6 [0.36–0.98] |
| | Stage E | 14 (48.3) | 2 (50.0) | 0.6, 0.63[0.07–5.53] | 22 (59.7) | 16 (38.1) | 0.004*, 0.53 [0.34–0.81] |
| | Stage F/G | 4 (13.8) | 1 (25.0) | | 4 (10.8) | 16 (38.1) | |
| Direction of the SPM long axis | Distal | 25 (86.2) | 3 (75.0) | 0.4, 0.43 [0.058–3.19] | 1 (2.7) | 0 (0.0) | 0.5, 0.43 [0.04–4.82] |
| | Perpendicular | 4 (13.8) | 1 (25.0) | | 6 (16.2) | 1 (2.4) | 0.1, 0.22 [0.03–1.37] |
| | Mesial | 0 (0.0) | 0 (0.0) | | 30 (81.1) | 41 (97.6) | |
| Total | | 29 (100) | 4 (100) | | 37 (100) | 42 (100) | |
| **Between premolars** | | | | | | | |
| Demirjian stage | Stage D | 10 (38.5) | 2 (28.6) | 0.35, 0.45 [0.09–2.37] | 14 (24.6) | 7 (31.8) | 0.16, 3.33 [0.78–14.17] |
| | Stage E | 13 (50) | 3(42.9) | 0.32, 0.47 [0.11–2.1] | 25 (43.9) | 13 (59.1) | 0.08, 3.42 [0.85–134.69] |
| | Stage F/G | 3 (11.5) | 2 (28.6) | | 18 (31.6) | 2 (9.1) | |
| Direction of the SPM long axis | Distal | 22 (84.6) | 6 (85.7) | 0.94, 1.07 [0.16–7.1] | 0 | 1 (4.5) | <0.001*, 3.74 [2.54–5.49] |
| | Perpendicular | 4 (15.4) | 1 (14.3) | | 5 (8.8) | 2 (9.1) | 0.92, 0.93 [0.25–3.3] |
| | Mesial | 0 | 0 | | 52 (91.2) | 19 (86.4) | |
| Total | | 26 (100) | 7 (100) | | 57 (10) | 22 (100) | |

(Continued)

| ABO-scores | | ABO score | | | | | |
|---|---|---|---|---|---|---|---|
| | | Maxillary | | | Mandibular | | |
| | | Score-0 n (%) | Score-1[R] n (%) | P value[c], RR (95% CI) | Score-0 n (%) | Score-1[R] n (%) | P value, RR (95% CI) |
| **III. Vertical dimension** | | | | | | | |
| **Marginal ridge between premolar and 2nd molar in extraction side** | | | | | | | |
| Demirjian stage | Stage D | 9 (42.9) | 3 (25) | 0.35, 0.45 [0.09–2.37] | 13 (28.9) | 8 (23.5) | 0.13, 0.63 [0.35–1.14] |
| | Stage E | 9 (42.9) | 7 (58.3) | 0.88, 1.09 [0.33–3.66] | 25 (55.6) | 13 (38.2) | 0.02*, 0.53 [0.31–0.91] |
| | Stage F/G | 3 (14.3) | 2 (16.7) | | 7 (15.6) | 13 (38.2) | |
| Direction of the SPM long axis | Distal | 18 (85.7) | 10 (83.3) | 0.69, 1.43 [0.24–8.38] | 1 (2.2) | 0 | 0.6, 0.52 [0.07–5.82] |
| | Perpendicular | 3 (14.3) | 2 (16.7) | | 7 (15.6) | 0 | 0.14, 0.13 [0.01–1.9] |
| | Mesial | 0 | 0 | | 37 (82.2) | 34 (100) | |
| Total | | 21 (100) | 12 (100) | | 45 (100) | 34 (100) | |
| **IV. Inclination[#]** | | | | | | | |
| **Bucco-lingual inclination of SPM[#]** | | | | | | | |
| Demirjian stage | Stage D | 8 (36.4) | 4 (36.4) | 0.28, 0.56 [0.19–1.63] | 1 (33.3) | 20 (26.3) | 0.32, 0.95 [0.87–1.05] |
| | Stage E | 12 (54.5) | 4 (36.4) | 0.12, 0.42 [0.14–1.26] | 2 (66.7) | 36 (47.4) | 0.16, 0.95 [0.88–1.02] |
| | Stage F/G | 2 (9.1) | 3 (27.3) | | 0 | 20 (26.3) | |
| Direction of the SPM long axis | Distal | 19 (86.4) | 9 (81.8) | 0.72, 0.8 [0.24–2.67] | 0 | 1 (1.3) | 0.16, 1.03 [0.99–1.07] |
| | Perpendicular | 3 (13.6) | 2 (18.2) | | 1 (33.3) | 6 (7.9) | 0.42, 0.88 [0.65–1.2] |
| | Mesial | 0 | 0 | | 2 (66.7) | 69 (90.8) | |
| Total | | 22 (100) | 11 (100) | | 3 (100) | 76 (100) | |
| **V. Root angulation** | | | | | | | |
| **Roots angulation of SPM[##]** | | | | | | | |
| Demirjian stage | Stage D | 8 (44.4) | 4 (26.7) | 0.79, 0.83 [0.22–3.18] | 2 (66.7) | 19 (25) | 0.15, 0.9 [0.79–1.04] |
| | Stage E | 7 (38.9) | 9 (60) | 0.56, 1.41 [0.44–4.47] | 1 (33.3) | 37 (48.7) | 0.3, 0.97 [0.92–1.03] |
| | Stage F/G | 3 (16.7) | 2 (13.3) | | 0 | 20 (26.3) | |
| Direction of the SPM long axis | Distal | 14 (77.8) | 14 (93.3) | 0.32, 2.5 [0.42–15] | 0 | 1 (1.3) | 0.15, 1.03 [0.99–1.07] |
| | Perpendicular | 4 (22.2) | 1 (6.7) | | 1 (33.3) | 6 (7.9) | 0.42, 0.88 [0.65–1.2] |
| | Mesial | 0 | 0 | | 2 (66.7) | 69 (90.8) | |
| Total | | 18 (100) | 15 (100) | | 3 (100) | 76 (100) | |

**Notes:**
SPM, Second Permanent Molar; ABO, American Board of Orthodontics; RR, Relative Risk.
[$] Score 0 for parameters <0.5 mm and 1 for parameters ≥0.5 mm.
[#] Buccolingual inclination of SPM: Score 0 for parameters <1 mm and 1 for ≥1 mm.
[##] Radiographic scoring for root angulation: score 0 for parallel roots, score 1 for nonparallel roots, or roots contacting adjacent roots.
[R] Reference.
[*] P value significant at 0.05.

among dentists regarding the management of severely decayed FPMs (*Meligy, 2016*). Extraction was reported to lead to several adverse consequences that could lead to occlusal disturbance or mispositioning of the teeth, thereby affecting oral health status (*Al Mansour et al., 2022*; *Ebrahimi et al., 2010*; *Saber et al., 2018*). Therefore, understanding the

**Table 4 Multilevel linear regression-analysis for the association between gender, arch of extracted FPM, Demirjian Stage of SPM and the direction of the long axis of SPM before FPM extraction (independent factors) and the mean combined ABO-cast measurement.**

| Variables | | Maxillary | | Mandibular | | Total | |
|---|---|---|---|---|---|---|---|
| | | B (95%CI) | P value | B (95% CI) | P value | B (95% CI) | P value |
| Gender | Male | −0.32 [−1.46 to 0.83] | 0.57 | 0.08 [−0.057 to 0.72] | 0.81 | −0.07 [−0.60 to 0.46] | 0.78 |
| | Female | 0.00 | | 0.00 | | 0.00 | |
| Demirjian stage | Stage D | −0.37 [−2.07 to 1.33] | 0.66 | −0.14 [−0.96 to 0.68] | 0.73 | −0.27 [−0.97 to 0.44] | 0.45 |
| | Stage E | 0.09 [−1.53 to 1.72] | 0.91 | −0.18 [−0.89 to 0.53] | 0.61 | −0.11 [−0.74 to 0.53] | 0.74 |
| | Stage F&G | 0.00 | | 0.00 | | 0.00 | |
| Direction of the long axis of SPM | Distal | −0.55 [−2.13 to 1.02] | 0.48 | −1.59 [−4.17 to 0.99] | 0.22 | −0.16 [−2.22 to −1.03] | <0.001* |
| | Perpendicular | 0.00 | | −0.95 [−2.01 to 0.11] | 0.08 | −0.94 [−1.77 to −0.12] | 0.025* |
| | Mesial | | | 0.00 | | 0.00 | |
| -2RLL | | 111.02 | | 256.7 | | 376.74 | |
| BIC | | 114.4 | | 261.0 | | 381.41 | |
| Covariance: estimate (SE) | | 2.15 (0.57) | | 1.62 (0.27) | | 1.71 | |

**Notes:**
FPM, First Permanent Molar; SPM, Second Permanent Molar; CI, Confidence Interval.
* Statistically significant at 0.05; -2RLL, -2 Restricted Log Likelihood; BIC, Schwarz's Bayesian Criterion; B, Estimate.

prognosis of extraction and its related predictors is important when considering elective extraction and effective treatment planning.

This study found that the frequency of having favorable pattern of spontaneous space closure was low. The direction of the SPM long-axis at pre-extraction time seems to be associated with more favorable results of ABO scores in the maxillary extraction sites compared to the mandibular. The stage of SPM development at the pre-extraction time was related to the ABO criteria of spacing between the SPM and second premolar and the vertical dimension in the mandible.

Previous studies on this topic lack reliable validated measurement tools and had many limitations in their methodology and design (*Saber et al., 2018*). Nevertheless, this study evaluated the spontaneous space closure pattern using clinical and radiographic parameters of the ABO grading system. This grading system was originally used to evaluate the successful completion of orthodontic treatment. Thus, this study is expected to provide solid evidence for assessing the favorable spontaneous outcomes of FPM extraction and the success of spontaneous space closure pattern in relation to occlusion and arch alignment conditions.

The study sample included more female than male patients and more mandibular teeth than maxillary teeth. This distribution in sex and tooth location was similar to a previous study conducted on FPM extractions (*Ozmen, 2019*). This could be related to the fact that the FPM erupts earlier in female patients and in the mandible than in male patients and the maxillary arch, which exposes them earlier to caries (*Fekonja, 2022*; *Javed et al., 2022*; *Sánchez-Pérez et al., 2019*).

Favorable space closure was seen in 6.58% of the total sample. This was lower than what was previously reported (more than 80% in the maxilla and 50% in the mandible) (*Ciftci*

*et al., 2021*; *Saber et al., 2018*). The wide differences are related to the diversity in the assessment tools used which could be considered comparable to only one of the ABO parameters which is spacing. Other parameters were not previously evaluated, making our study's assessment more comprehensive.

Previous studies have assessed space closure using either radiograph (*Telli & Aytan, 1989*), or clinical measurements (*Teo, Ashley & Derrick, 2016*). However, clinical measurements was reported to have moderate to low reliability compared to cast measurements when evaluating rotation, malocclusion severity, spacing, and inclination (*Ovsenik, Farcnik & Verdenik, 2004*). Moreover, the precision of these techniques can be affected by factors such as radiograph positioning and accuracy (*Hegde et al., 2023*). However, this study used orthodontic cast which is considered the current gold-standard reference for occlusal assessment in orthodontics (*Rheude et al., 2005*).

Our findings indicated that space closure between the SPM and premolar teeth was the most favorable outcome among all other ABO parameters, consistent with previous reports. This space closure may explain why many patients do not notice the missing FPM, do not experience discomfort with their teeth, and do not seek orthodontic treatment later in life (*Demir & Aydoğdu, 2020*).

In our sample, most FPMs were extracted at a younger age and at Demirjian stages D and E, which could be attributed to the general belief that the main factor for improving the pattern of extraction space closure is the early time of extraction, and when the SPM is in the early stage of development (*Saber et al., 2018*). This is also supported by our finding that the early SPM development stage increased the probability of space closure between the SPM and second premolar and favorable vertical dimension in the mandible. Nevertheless, previous studies have primarily focused on space closure and third molar position, rather than ABO parameters (*Noar et al., 2023*).

Moreover, distal long axis inclination of the SPM, which accounted for 84.8% of SPM in the maxilla, increased the probability of favorable ABO scores in all the parameters. Therefore, the extraction of a young maxillary FPM is not expected to affect the occlusion and alignment of the arch.

This study aimed to explore different aspects of spontaneous space closure following FPM extraction. It assisted the key indicators for an optimal pattern of spontaneous space closure and alignment after early FPM extraction. However, here are some limitations. The risk of selection bias and the limited number of extractions in certain subgroups, such as older patients (aged >10 years) and in the maxillary arch indicates the need for future studies including a large sample from the maxilla that equals that of the mandible. Nonetheless, the generalizability of this study was enhanced by involving three major referral healthcare centers that serve diverse populations from various community groups. Furthermore, the study included a sample that closely resembled the demographic distribution of children who undergo the extraction of FPM.

## CONCLUSIONS

This study provides a comprehensive analysis of spontaneous space closure patterns of the FPM extraction space, using the ABO grading system. While acknowledging the

significance of the direction of the SPM long axis and the early Demirjian stage of SPM development as valuable parameters for enhancing favorable pattern of space closure in extraction sites, it is still anticipated that orthodontic treatment will be needed in most cases of compromised FPM extraction. Furthermore, our findings indicate that achieving a favorable score of spacing is the one ABO-parameter that is predominantly observed after FPM extraction compared to the other commonly overlooked, yet relatively important parameters.

### Funding
The authors received no funding for this work.

### Competing Interests
The authors declare that they have no competing interests.

### Author Contributions
- Heba Jafar Sabbagh conceived and designed the experiments, analyzed the data, prepared figures and/or tables, authored or reviewed drafts of the article, and approved the final draft.
- Ahmad Adnan Samara performed the experiments, authored or reviewed drafts of the article, and approved the final draft.
- Shoroog Hassan Agou conceived and designed the experiments, authored or reviewed drafts of the article, and approved the final draft.
- Jihan Turkistani performed the experiments, authored or reviewed drafts of the article, and approved the final draft.
- Manal Ibrahim Al Malik performed the experiments, authored or reviewed drafts of the article, and approved the final draft.
- Hadeel Abdulatief Alotaibi performed the experiments, authored or reviewed drafts of the article, and approved the final draft.
- Abrar Shimi Dhaifallah Alsolami performed the experiments, authored or reviewed drafts of the article, and approved the final draft.
- Nada Othman Bamashmous conceived and designed the experiments, analyzed the data, prepared figures and/or tables, authored or reviewed drafts of the article, and approved the final draft.

### Human Ethics
The following information was supplied relating to ethical approvals (*i.e.*, approving body and any reference numbers):

Ethical approval was obtained from the Research Ethics Committee of King Abdulaziz University Faculty of Dentistry (#005-01-19), Research Ethics Committee of King Fahad Armed Forces Hospital (#REC 277), and the Institutional Review Board (IRB) of Ministry of National Guard at King Abdullah International Medical Research Center (#H-01-R-005).

## Data Availability

The data used in this study is available in the Supplemental File.

## Supplemental Information

Supplemental information for this article can be found online at http://dx.doi.org/10.7717/peerj.18276#supplemental-information.

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
