# Peer review of "Spontaneous space closure after extraction of young first permanent molar. Retrospective cohort study"

_PeerJ, doi:10.7717/peerj.18276_

## Round 0.1 · original submission · Major Revisions

Consider primarily the comments of reviewers 1 and 3 for the revision. Among other comments, the Introduction needs to be strengthened.

Note that the Materials and methods section needs to report which are the tested outcomes.

Please retain the term sex instead of gender.

·

Basic reporting

Thank you for sending me for review the entitled ''Spontaneous space closure after extraction of young first permanent molar. Retrospective cohort study'' The manuscript was evaluated and some comments are below:
Abstract
1- ''Dimerjian stage D/E'' please check the writing
2- ''optimal pattern of spontaneous space closure after FPM early extraction'' optimal pattern should be '' favorable outcome, please check the usage of optimal pattern
3-
Introduction
1-'' American Board of Orthodontics (ABO) cast-radiograph model grading system'' This sentences need to use a reference, please check
2- the authors used e new evaluation tool-method-cirteria in this article for Spontaneous space closure after extraction of the first permanent molar. This sounds good but I think the authors should mention the previous study and their limitations ın the introduction part of the manuscript.
Material and Methods
1-line 117, samples are not necessary
2-data collection tool part is non-frame, please check and simplify maybe hesitate commans and separate sentences
3- Figure 1 should present a combined figure. After the Figure 1 presentation, A, B,C should be clarified below the title of the figure
4-Figure 2 should present a combined figure. After the Figure 2 presentation, A,B, C should be clarified below the title of the figure
5- Figures 3A, B,C, D low quality, please check, especially, figure 3D is opposite shown
6- line 198, this paraph needs a title, etc statistical analysis

Results
1- table 1, it's not necessary UDH n (%) KFAFH n (%) KAMC, because the authors mentioned this in the material method section
2- I think the no of extracted teeth (16,26,36,46) should separate as mandibular and maxillar, please consider this
3- ıf ıt possible tables should be need simplification. please check this

Discussion
1- line 292, I think the ıntro of discussion should not start this type of sentence'' The current study showed the pattern of spontaneous space'' The authors should maybe use this on another part of the discussion
2- line 305, Previous studies were written but only one reference article was used, please check
3- line 319, please check the sequences of reference articles in this data of sentences
4- Generally, the discussion is short, maybe the authors should mention previous studies and discuss their results

Experimental design

Material methods look good and well designed, Sample size is small but the authors have mentioned the limitations of manuscript.

Validity of the findings

tables with findings are mixed, the comments about this are present

Additional comments

''Spontaneous space closure after extraction of young first permanent molar. Retrospective cohort study'' very interesting and important issue, I think Articles covering such topics will be interesting and useful for clinicians.

·

Basic reporting

This is a well written and researched article that investigates an important topic in a systematic and clear manner. The references quoted are relevant but like most works there are a few that have been missed. Having said that I don't think there are references omitted that would challenge the authors assertions. The tables are well constructed and clear and the photographs clear and helpful
The results are well presented and the conclusions adequately represent the findings. I think the authors of this study should be congratulated for their submission.

Experimental design

I agree that this work is well presented with the research question well defined, relevant & meaningful.
The management of FPM's of poor prognosis is becoming more important particularly with the increase in prevalence of MIH. The investigation is presented and was performed to a high technical & ethical standard. The methods were described with excellent detail & information.

Validity of the findings

Again in my opinion the data has been well presented appear robust, statistically sound, & controlled.

Conclusions are well stated, linked to original research question & limited to supporting results.

Additional comments

Well done for a well designed, implemented and reported piece of work

·

Basic reporting

General comments:
• I suggest that the authors replace the words “children” and “participants” with “patient’s records” or “patients” in the whole document. The article need to be a united unity, otherwise the readers lose the common thread.
Abstract:
• The background section is too short and need to be longer.
• In the method section, at the end of second line, I suggest that the authors write that the patient’s records were RETROSPECTIVELY screened.
• In the method section, at the middle of the fourth line, the authors used these parentheses [ ] instead of these ( ). These parentheses ( ) were used otherwise in the whole document. It would be beneficial to use the same type all along the document.
• In the method section I recommend to not mention what variables were studies, so the background section can be made longer
• The term “spontaneous space closure” was written twice, one time with a hyphen between the words “Space” and “closure” and the other time without. I suggest that the hyphen would be removed.
Introduction: Very good written
• Line 68, the word “an” before the “individual” is not needed.
• Line 77, the reference is missing after the word methodology.

Experimental design

Materials and methods: Excellent work to follow the STROBE guidelines.
• From line 102 to line 111, the inclusion and exclusion criteria would be easier to be read if they were written point by point as done in the data collection section.
• From line 102 to line 105, in the inclusion criteria it would be beneficial to add: The presence of radiological images at the time of the extraction as an inclusion criteria. It would also be positive to mention what type of radiological images were searched and what teeth needed to be presented to assume that the radiological image is eligible.
• Line 104, I suggest to replace the word “and” with “or”.
• Line 115 the word “participants” was used, but in line 119 the word patients was used in the same context. I would recommend to use the same word in the whole document to make it as a united unity.
• The information in the end of line 125 until the middle of line 127 is a repetition of the information written in the lines above.
• Lines 136 and 137, the direction of SPM was divided into three catigories, but the tool to decide when the tooth was perpendicular was not mentioned. Was it just visualization or any type of measuring instrument?
• Line 176, it would be easier for the reader to replace the word “treatment” by “extraction”.
• At the end of line 176 and line 177, I would recommend to explain more about the patient consent. Did the patients for example consent to publish their radiological images?
• The section for Ascertainment and reliability, it is good to write the initials of the people who participated in the work.

Validity of the findings

Results:
• Line 218, the word “the” need to be deleted.
• Lines 219 and 220, as the authors didn’t obtain a consent from all the 192 patients, I am not sure if it would be right to represent any demographic information about the group as the amount of female in it.
• The authors are describing tables 3 and 4 in the 6:th and 7:th paragraph. The result section need to be a presentation to the raw data as done in the paragraphs before and after. I would recommend to restructure these paragraphs so they will be presentative and not descriptive as they are now.
• Line 253, I believe that the authors mean “extraction location” instead of just location.
• Line 253, I believe that the authors mean “SPM direction” instead of position.
• Line 253, I believe that the authors mean “developmental stage” instead of development.
• Line 253, the authors wrote the abbreviation RR without defining it before.
• From the end of the line 253 to line 258, I suggest that the authors begins this whole part with -In maxillary FPMs extractions- then continue to talk about the results. Exactly as done in line 258. This clarify in a better way that all the data described after is about the maxillary extractions.
Discussion:
• The first paragraph is interesting and I suggest the authors to end it with –Which was an interesting fining in this study- because it was not discussed in the literature earlier.
• I personally think that lines from 300 to 307 at the middle, belongs to the introduction section.
• Line 308, no hyphen between the words “space” and “closure” is needed.
• Line 313, at the middle, please use “gender” instead of “sex”
• I would recommend to mention the risk of selection bias in the limitation paragraph as it is a general limitation in the retrospective cohort studies.

Additional comments

Tables:
• In table 1, I recommend to replace the word “Sex” by “Gender”
• In table 1, the column for P-value is insignificant.
• In table 1, the cell for “No. of extracted FPMs/ patients” I assume that it has to be just “No. of extracted FPMs”.
• In table 2, in the first line under the table. The authors described that CI is confidence interval but I couldn’t find it in the table.

• In table 3, in the last two lines under the table. The authors mentioned twice that the P value is significant at 0.05
• In table 3, the * used to highlight the significant values is used sometimes before the value and sometimes after. The authors need to decide where it will be placed and do the same to all values. Example: The rotation with the distal direction of the SPM in the maxillary extractions, compared with the spacing between SPM and second premolar with stage D in mandibular extractions.
• The P-value for the spacing between SPM and second premolar with stage E in mandibular extractions is significant but no * was used.
• The same as the point above for the spacing between premolars with distal direction of the SPM in mandibular extractions
• All tables don’t follow the same format .this needs to be adjusted.
Figures:
• I would recommend to change all the subfigures under figure 2 with the original figure in Demirjian’s article.

---

## Round 0.2 · Minor Revisions

The reviewers were satisfied with the revisions, but they requested a few additional very minor changes to improve the reporting. Therefore, I recommend proceeding with these amendments and resubmitting your manuscript for publication.

·

Basic reporting

Abstract
-results: For the mandibular arch, ...Dimerjian stage D and E significantly... please correct write Demirjian instead of Dimerjian.

All previous comments and corrections were done by Authors. Thank you

Experimental design

I agree that this paper is well presented with the research question well defined, relevant &
meaningful. A new diagnostic post-measurement tool (ABO) is different from previous studies.

Validity of the findings

Data has been well presented appears robust, and is statistically sound. This a cohort study and the findings are adequate.

Additional comments

Well done for a well-designed, implemented and reported piece of work

·

Basic reporting

No comments

Experimental design

• Line 115, please change the term ”included criteria” to “inclusion criteria”.
• From line 116 to line 130, please use capital letters in the first letter of first words after the points.

Validity of the findings

• Line 279, the sentence “extracting the FPM in the maxillary arch” is a repetition to the sentence before and need to be deleted.

Additional comments

No comments

---

## Round 0.3 · Minor Revisions

All reviewers have been satisfied with the revisions, but there are two more issues that need to be addressed.
1. According to the journal guidelines, multi-part figures should be submitted in one file and arranged as they are to be published. Also, titles are required for all figures. Please follow the journal guidelines (https://peerj.com/about/author-instructions/#figures) for Figure submission.
2. The typo "Dimerjian" appears 8 times in the Results section. Please correct it throughout the manuscript.

---

## Round 0.4 · accepted · Accept

The authors have addressed all reviewers' and editor's comments adequately. The manuscript is now ready for publication.